# The influence of the age-period-cohort effects on the temporal trend mortality from schistosomiasis in Brazil from 1980 to 2014

**Taynãna César Simões** [1]*, **Roberto Sena** [1], **Karina Cardoso Meira** [2]

1 Oswaldo Cruz Foundation—Fiocruz, René Rachou Institute, Belo Horizonte, Minas Gerais, Brazil,
2 School of Health, Federal University of Rio Grande do Norte, Rio Grande do Norte, Brazil

☯ These authors contributed equally to this work.
* taynana.simoes@fiocruz.br

## Abstract

### Background

Schistosomiasis is highly debilitating and related to poverty, leading to chronic health problems. This disease is important for public health due to the high prevalence, severity of clinical forms and heterogeneous spatial and temporal patterns. In Brazil, about 1.5 million people are at risk of infection with *Schistosoma mansoni*, with an annual average of 500 deaths. In this study, the temporal change in mortality was evaluated in relation to the effects of age, period and birth cohort, in Brazil and regions, from 1980 to 2014.

### Methods

In this study, we analyzed the influence of age, period and birth cohort (APC effects) on the temporal evolution of schistosomiasis mortality in Brazil from 1980 to 2014, according to sex and geographic regions of the country. The death records were extracted from the SIM (Mortality Information System) of the DATASUS website (Department of National Health Informatics) of the Ministry of Health of Brazil. The temporal effects were estimated using Bayesian models and the INLA (Integrated Nested Laplace Approximations) method for parameter inference.

### Results

More than 24 thousand deaths were registered in the analyzed period, mainly in men from the Northeast region. In Brazil, children under 14 years of age had protection against death from schistosomiasis. There was no significant effect for others ages. From 1990 to 1999, there was a protective effect for death from schistosomiasis and a null effect in the other periods. There was a decreasing trend in the risk of death among birth cohorts. The greatest risk was among people born from 1903 to 1912. There was a protective effect for death among people born after 1968. Men were at risk of death between 25 and 54 years old, while women were at risk after seventy years of age. The southern and central-western regions had a risk of death until 1989 and had a protective effect between 1995 and 1999.

(https://datasus.saude.gov.br/informacoes-de-saude-tabnet/).

**Funding:** This article is a product of a project coordinated by the main author and funded by the National Scientific and Technological Development (CNPQ). The funders had no role in study design, data collection and analysis, decision to publish, or preparation of the manuscript. http://www.cnpq.br/

**Competing interests:** The authors have declared that no competing interests exist.

The northern region had a risk of death between 1985 and 1994, and a protective effect after 2005. The northeast and Southeast regions had protective effects for death between the years 1990 and 1999, and after 2000, respectively. People born until 1952 and 1957 were at risk of dying in the South and North regions, respectively, and a protective effect among people born after 1968, in both regions. In the Northeast region, there was a protective effect among people born after 1963. In the other regions, there was a risk of death among people born until 1962 and a protective effect among people born after 1973. The Central-West region had the least declining trend in risk of death among birth cohorts.

## Conclusions

The birth cohorts had a great influence on the decreasing trend of schistosomiasis mortality in Brazil. This result may be due to the interaction between demographic changes and greater access to health and sanitation services, in addition to the impact of schistosomiasis control measures experienced by younger cohorts.

## Introduction

*Schistosomiasis mansoni* is an acute and chronic disease caused by parasitic worms of the genus *Schistosoma*. The transmission of the disease depends on the presence of the infected man, excreting helminths eggs through feces. Intermediate hosts are aquatic snails that release infective larvae from the worm into the water collections used by humans. It is a tropical disease, registered in 54 countries, mainly in Africa and the eastern Mediterranean, affecting the regions of the Nile Delta and countries like Egypt and Sudan. It occurs in South America, especially in the Caribbean, Venezuela and Brazil. In Brazil, it is estimated that 1.5 million people are at risk, in all geographic regions, with the states of the Northeast and Southeast regions being the most affected. Between 2006 and 2015, there were approximately 508 annual deaths in the country [1,2].

The disease is more prevalent in poor communities, without access to clean water and adequate sanitation. Other factors, such as socioeconomic status, occupation, leisure, educational level and information of the population exposed to the risk of disease risk, contribute to the occurrence of schistosomiasis in a locality. Although clinically diverse, neglected tropical diseases share characteristics that allow them to persist in conditions of poverty, where they are grouped and overlap. In this context, WHO has considered schistosomiasis as an integrated approach to control neglected tropical diseases [1,3–5].

In Brazil, the actions of the Epidemiological Surveillance of Schistosomiasis carried out by the Ministry of Health allow the diagnosis and treatment of patients with *S. mansoni*, in order to: a) reduce the occurrence of severe forms and, consequently, of deaths; b) reduce the prevalence of infection; c) indicate measures to reduce the risk of disease spread [6]. Since the introduction of new drugs administered in a single dose and their use on a large scale, a significant reduction in the number of cases that developed severe forms of schistosomiasis has been observed [7]. Nowadays, the lethality is considered low. However, even with such measures, between 2006 and 2015 there were about 508 deaths per year in the country [1,2]. So, surveillance of the magnitude and dynamics of transmission, severe forms and deaths related to schistosomiasis in Brazil are essential to monitor and evaluate the effectiveness of disease control measures [7–10].

The continuous analysis of the evolution of health indicators, especially mortality rates, is important to provide information that supports the development of specific public policies. In particular, this change in time can be influenced by three temporal factors: age, period and birth cohort. The effects of age (A) may be related to physiological factors, which cause changes in the individual with increasing age. The effects of the period (P) can be related to changes caused by events that occur at specific moments in time, simultaneously influencing all age groups, such as social and epidemiological factors. In turn, cohort effects (C) are related to factors that affect an entire generation, such as individuals who were born in the same year, and therefore have similar habits and behaviors. The cohort effects may be due to the accumulation of exposures throughout life, which cause changes of different magnitudes in successive age groups and time intervals. Therefore, these effects can also be understood as resulting from an interaction between the effects of age and period [11,12].

The aim of this study was to analyze the effects of age, period and birth cohort on the temporal evolution of mortality from schistosomiasis in Brazil, according to sex and geographic regions, in the last three decades, considering cohorts from 1898–1902. Unstructured random terms were evaluated in order to incorporate over-dispersion in the data and unobservable factors.

## Methods

### Ethics statement

This study analyzed secondary data of free access, available for download on the website of DATASUS–Department of Informatics of the Unified Health System of Ministry Health of Brazil. The smallest geographic unit available is the city level, so it is impossible to identify the individual, guaranteeing their anonymity. Thus, it was not necessary to submit this study to the local ethics committee.

### Study design and data source

This is a population-based longitudinal ecological study to analyze the time trend of schistosomiasis mortality rates in Brazil. The country is divided into 26 states and a federal government grouped into five major regions (North, Northeast, Southeast, South and Central-West) with different geographical, economic and cultural characteristics. The Southeast (88,072,407 inhabitants) is the most populous and economically most developed region, and the South region (30,036,030 inhabitants) has the best human development indexes. On the other hand, the Central-West (16,293,774 inhabitants) has an economy based on agriculture and livestock, the Northeast (57,883,049 inhabitants) has the lowest human development rates and the North (18,373,753 inhabitants), which includes the Amazon rainforest, has low population density and ranks second with the lowest human development rates. Population data were obtained from the IBGE website (Brazilian Institute of Geography and Statistics).

The annual death records between 1980 and, specific for age, were extracted from the Mortality Information System of the SUS (Unified Health System) Department of Information Technology (SIM-DATASUS) of the Ministry of Health of Brazil (DATASUS, 2017). The categories of underlying deaths selected in the SIM were Schistosomiasis—codes 120 (ICD-9 until 1995) and B65 (ICD-10 after 1996). SIM was created for the regular collection of data on mortality in the country, allowing the collection of data in a comprehensive way, in order to subsidize the different spheres of management in public health. Death certificates contain demographic (age, gender, education, race, marital status, municipality of residence and municipality of occurrence of death) and clinical information (underlying and associated causes of death). It is the physicians' responsibility to complete the death certificates. Until

1995, reference codes were based on the International Classification of Diseases (ICD) in its 9th revision, and after 1996, they were based on the 10th revision of the ICD. All data were collected in November 2017 [13].

The last two years available in the system (2015–2016) were discarded in the Age-Period-Cohort (APC) analysis, in order to obtain five-year intervals for age and period, resulting in I = 17 age groups (less than 4 years up to 80 years old or more), J = 7 periods (from 1980 to 2014), and K = I + J—1 = 23 birth cohorts (from 1898 to 2012). In addition to data for Brazil as a whole, we collected deaths from schistosomiasis according sex and geographic regions.

The procedure for correcting death records and standardizing rates were performed using all age groups. Standardization was done according to the distribution of specific rates by age and sex, using the direct method, based on the Brazilian population of the 2010 Census, computed for 100,000 inhabitants. It was decided to correct the records due to the great disparity in the quality of information between Brazilian geographic regions. In addition, during the analysis period, there were changes in the death registry in the country, with the implementation of SUS [5]. Corrected death records were added to observed deaths, following two steps: (1) proportional redistribution of records classified with ignored age and/or sex among deaths classified as schistosomiasis as the underlying cause; (2) treatment of deaths classified as by ill-defined causes using ICD-9 codes 780–799 and ICD-10 codes R00-R99, according to year and age group [14].

## Modeling procedure

Many proposals using classical approaches for the Age-Period-Cohort models (APC models) attempted to solve the well-known problem of non-identifiability resulting from the linear relationship between the three temporal factors. However, there is no consensus or definitive solution in the literature [15–20]. Under a Bayesian approach, the temporal effects in the APC models (age, period and cohort) are related to rates according to prior smooth functions. In addition, it is possible to incorporate random effects and additional structures in time and space, in a less complex way. The hierarchical model incorporates uncertainty into hyperparameters (parameters of prior distributions), bypassing the problem of non-identifiability. Another advantage of using the Bayesian approach is the possibility of statistically testing the differences in the estimated effects between the control variables, such as sex and geographic regions [21,22].

In the modeling procedure, a first univariate Bayesian APC model was adjusted in order to estimate the temporal effects on the risk of death from schistosomiasis for the whole of Brazil [23]. Then, multivariate models were adjusted [21], in which $y_{ijkg}$ and $n_{ijkg}$ are deaths and people at risk in the age group $i = 1,\ldots,17$, period $j = 1,\ldots,7$, and sex or region $g$. As an example of

**Table 1. Multivariate models considering different temporal and random effects, according to sex (a) or geographic region (b).**

| APC Models | Linear Predictor |
|---|---|
| Model 1a (1b) | Join effects for men and women (or geographic regions) |
| Model 2a (2b) | Age-specific effects by sex (or geographic regions) |
| Model 3a (3b) | Period-specific effects by sex (or geographic regions) |
| Model 4a (4b) | Cohort-specific effects by sex (or geographic regions) |
| Model 5a (5b) | Age-specific and period-specific effects by sex (or geographic regions) |
| Model 6a (6b) | Age-specific and cohort-specific effects by sex (or geographic regions) |
| Model 7a (7b) | Period-specific and cohort-specific effects by sex (or geographic regions) |

one of the models used, considering the period effects as constants, we have:

$$y_{ijkg} \sim Poisson(n_{ijkg}exp(\varepsilon_{ijkg}))$$
$$\varepsilon_{ijkg} = \mu_g + \alpha_{ig} + \beta_j + \gamma_{kg} + z_{ijkg}$$

(1)

where, $\mu_g$ is the specific global average by sex/region $g$, $\alpha_{ig}$ is the effect of age group $i$ according sex/region $g$, $\beta_j$ is the effect of period, $\gamma_{kg}$ is the effect of birth cohort according sex/region $g$, and $z_{ijkg} \sim N(0,\delta^{-1})$ are the random effects that consider the extra variability in $y_{ijkg}$ (over-dispersion) [23]. The other adjusted models are shown in Table 1.

To achieve the identification of the intercept term $\mu$, the constraint $\sum_{i=1}^{I} \alpha_i = \sum_{j=1}^{J} \beta_j = \sum_{k=1}^{K} \gamma_k = 0$ is assigned for each sex/region category. Under the Bayesian perspective, the problem of non-identifiability of latent parameters remains, however, additional restrictions are not required, unlike classical models. The temporal effects have received first-order random-walk (RW1) as prior distributions, uniform prior is assigned to $\mu$ and non-informative priors to the precision parameters ($G(1,0.00005)$).

The INLA (Integrated Nested Laplace Approximations) method was used to infer parameters. INLA is an alternative method to MCMC (Monte Carlo Markov Chain) for latent Gaussian models [24]. The Deviance Information Criterion (DIC) and the log-score were used to compare models. Both have a negative meaning, in the sense that the lower the values, the more appropriate the model is [25,26]. All analyzes were performed using the statistical software R, with the INLA package [27].

## Results

### Describing the data

In Brazil, 19,988 deaths whose underlying cause was schistosomiasis were reported underlying during the period analyzed (1980–2014). After correcting to ill-defined causes, in addition to ignored age and sex, 24,055 deaths from schistosomiasis were computed in the period considered.

The crude death rate from schistosomiasis was 2.99 deaths per 100,000 inhabitants during the entire period analyzed (1980–2014). The standardized annual average rate was 0.45 deaths per 100,000 inhabitants. Fig 1 shows the distribution of crude schistosomiasis mortality rates specific for age (a and d), period (b and e), and birth cohort (c and f), and according sex (a-c), and geographic region of residence (d-f). Mortality rates are highest among men and in the Northeast. The standardized annual mortality rate in men was 0.53 deaths per 100,000 men, and in women 0.37 deaths per 100,000 women. The standardized annual mortality rates in the Brazilian regions were 1.28, 0.32, 0.14, 0.09 and 0.05 deaths per 100,000 inhabitants, in the Northeast, Southeast, Central-West, North and South regions, respectively.

Rates increased with increasing age, decreased over the years, and were higher for older generations (birth cohorts) (Fig 2). However, the effects of these temporal terms seem to differ between men and women, and between Brazilian geographical regions.

### APC effects on mortality in Brazil

According to the univariate Bayesian model adjusted for the whole of Brazil, the estimated risk of death from schistosomiasis according to age ranged from 0.34 to 1.94, with a protective effect against death in children under 14 years of age, and zero risk for others age (Fig 2A). The risk of death according to the period varied between 0.85 and 1.26, with a protective effect against death between 1990 and 1999. The risk was null in the other periods observed (Fig 2B). The risks of death from schistosomiasis, according to the birth cohorts, were the most

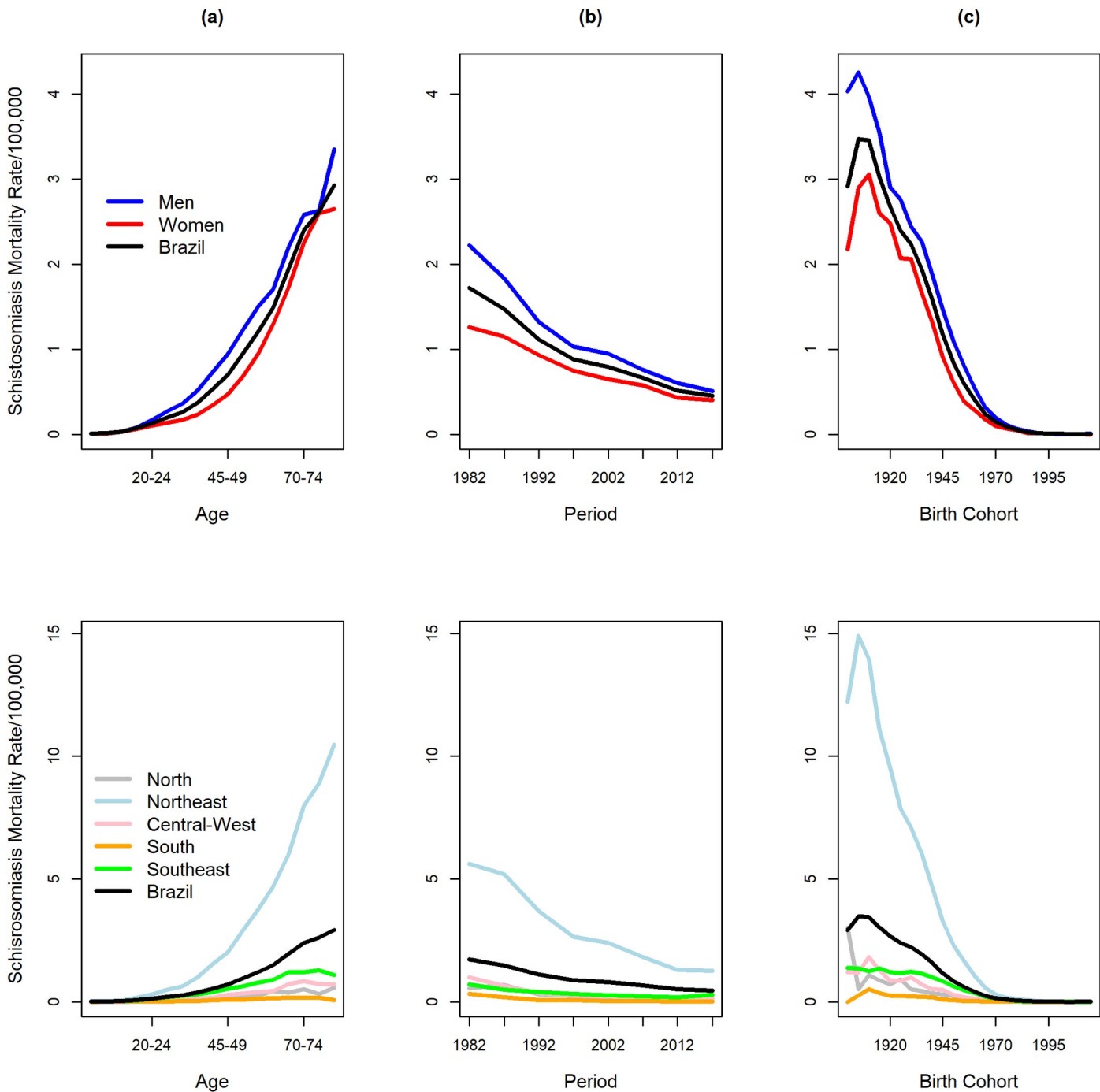

**Fig 1.** Distribution of observed schistosomiasis mortality rates according to age (a), period (b) and birth cohort (c), and according to sex and Brazilian regions, Brazil, 1980–2016.

significant, ranging from 0.01 to 17.61, and decreasing exponentially from the older cohorts to the younger ones. The risk of death was greater for people born between 1903 and 1912, and there was a protective effect for death among people born after 1968 (Fig 2C). The random effects that incorporated data over-dispersion, representing unobserved variables, presented a risk of death ranging from 0.73 to 1.36.

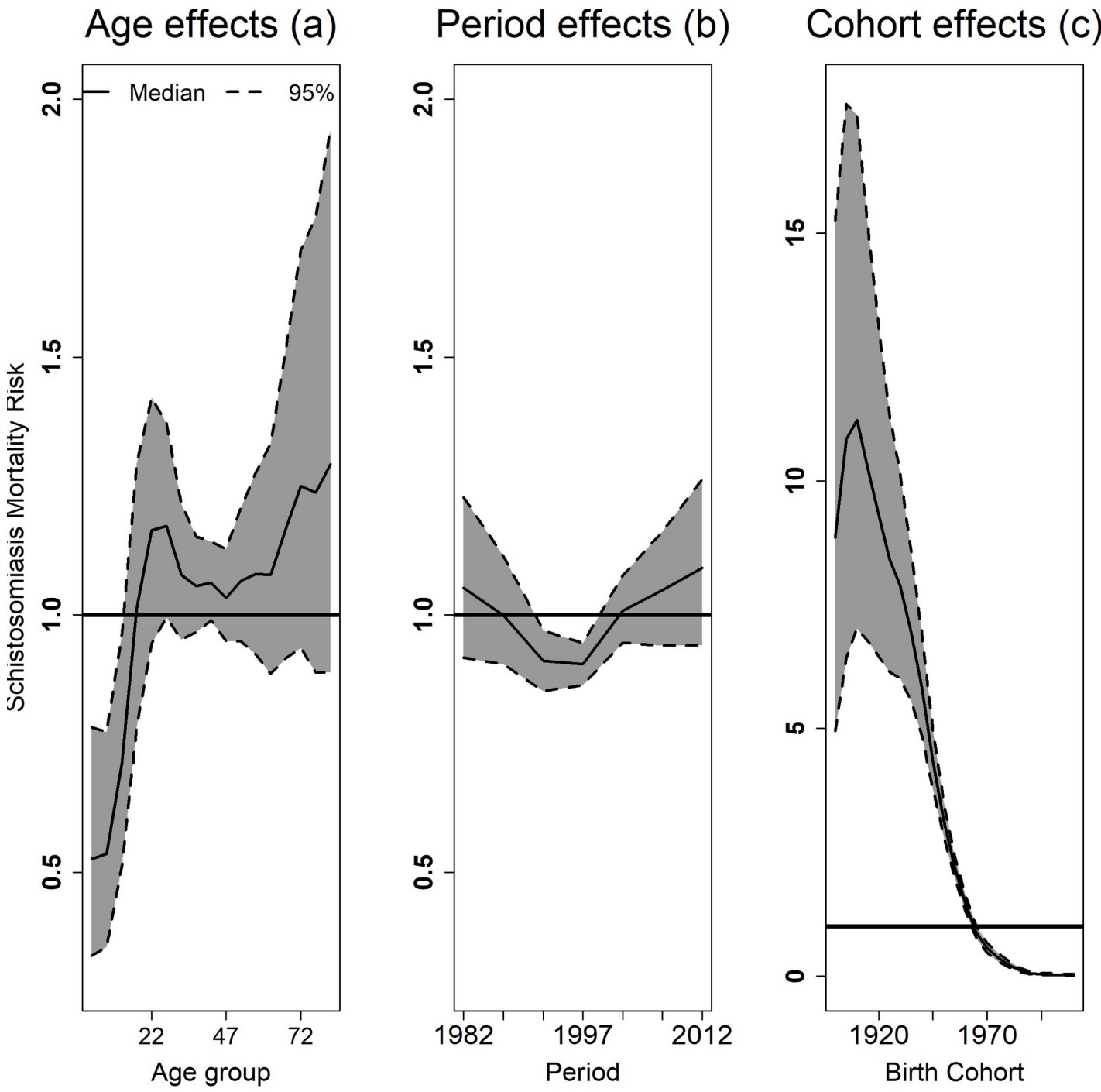

**Fig 2.** Estimated risks of schistosomiasis mortality and 95% credibility intervals according to the effects of age (a), period (b), birth cohort (c), Brazil, 1980–2014.

The Bayesian multivariate models (BAPC) were more significant than the univariate models, stratified according to sex and geographic regions (Table 1). Based on the results of Table 2, according to sex (models · a), the best adjustment was that with different age effects between men and women (model 2a). Among the multivariate models according to Brazilian regions (models · b), the best fit model considered differences in the period and cohort effects.

**Table 2. Comparison of models adjusted with temporal terms and random effects, according to sex and geographic regions (priors RW1 for temporal effects).**

| Evaluating APC effects according to sex (Selected **Model 2a**) | | | | | | | |
|---|---|---|---|---|---|---|---|
| Model | 1a | 2a | 3a | 4a | 5a | 6a | 7a |
| Log Score | 4.08 | 3.82 | 4.10 | 4.00 | 3.83 | 3.84 | 4.00 |
| DIC | 422.18 | 386.45 | 424.86 | 412.73 | 387.96 | 388.27 | 418.45 |
| Evaluating APC effects according to Brazilian Regions (Selected **Model 7b**) | | | | | | | |
| Model | 1b | 2b | 3b | 4b | 5b | 6b | 7b |
| Log Score | 2.78 | 2.76 | 2.72 | 2.71 | 2.59 | 2.65 | 2.59 |
| DIC | 939.90 | 980.94 | 894.51 | 968.17 | 890.50 | 925.30 | 877.11 |

Estimates of the risk of death from schistosomiasis, according to sex, showed that, although men had a protective effect against death from the age of 14, they presented a significant risk between 25 and 54 years. While women had a protective effect against death between 30 and 45 years, and significant risk after seventy years of age (Fig 3).

## APC effects on mortality in regions

Fig (4A and 4B) shows the differences in the risk of death from schistosomiasis between Brazilian regions, according to the effects of period and birth cohort. The South and Central-West regions were at risk of death until 1989 and a subsequent protective effect between 1995 and 1999. The North region had a high risk of death between 1985 and 1994, with a protective effect after 2005. The Northeast and Southeast regions had an effect protective against death after 1990, and zero risk in other periods. People born until 1952 and 1957 were at risk of death in the South and North regions, respectively, but people born after 1968 showed protection against death in both regions. In other regions, there was a risk of death among people born until 1962, and protection against death among people born after 1973, with the exception of the Northeast, which had a protective effect against death among people born after 1963. The Central-West region had the slowest decline in the risk of death from schistosomiasis over the generations.

## Discussion

In Brazil, *Schistosomiasis mansoni* is endemic in a large area and considered a serious public health problem, as it affects millions of people, causing a significant number of severe forms and annual deaths [28]. In this study, it was observed that the death rates decreased over the period of time analyzed, increased with increasing individual's age, and were higher in the older birth cohorts. The Age-Period-Cohort analysis showed a protective effect against death from schistosomiasis in children under 14 years of age. Men were at significant risk of death between 25 and 54 years old, while women were at risk after 70 years of age.

In the present study, there was a protective effect against death between the years 1990 and 1999, and zero risk in the other periods analyzed. There were differences in period effects between geographic regions. The South and Central-west regions were at risk of death until 1989, with a protective effect between 1995 and 1999. The North region was at risk between 1985 and 1994, and a protective effect after 2005. The Northeast and Southeast regions had a protective effect between 1990 and 1999, and up to 2000, respectively. Zero risk in other periods.

Our findings reinforce the pattern of decline observed in previous studies, which considered death from schistosomiasis as the underlying cause [7–9,29]. Official data from health information systems show a decrease in the number of deaths and hospitalizations for this

## Age Effects

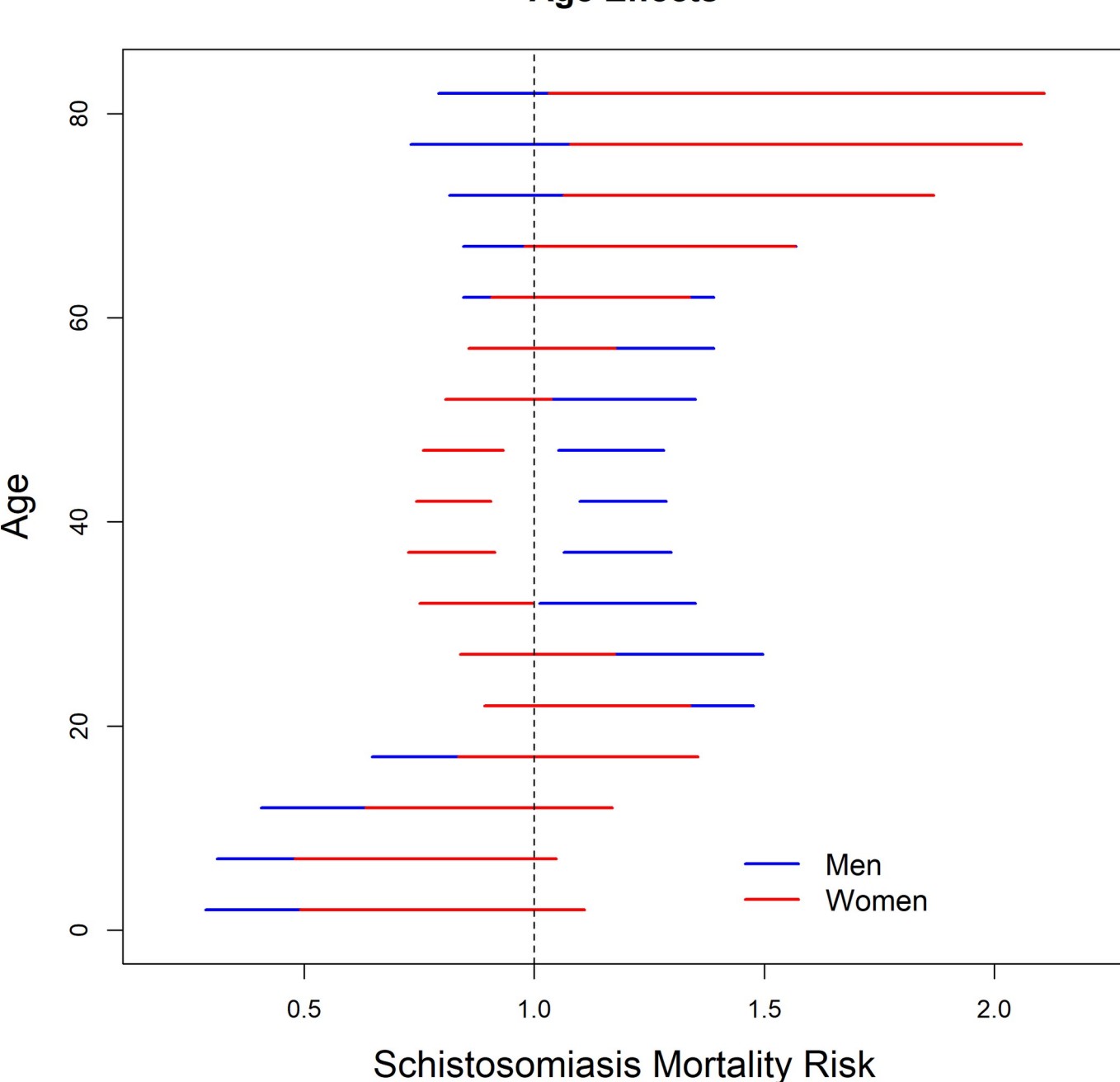

**Fig 3. 95% Credibility intervals for the estimated risk of death from schistosomiasis, specific for age, according to sex, Brazil, 1980–2014.**

disease in Brazil [30–32]. Between 1977 and 2003, Amaral et al. (2006) found a significant reduction in the percentage of people infected, as well as in hospitalizations and mortality from the disease [9]. Martins-Melo et al. (2014), in a national population-based study, found a decrease in mortality from schistosomiasis in the country, with different patterns between geographic regions, sex and age groups [30].

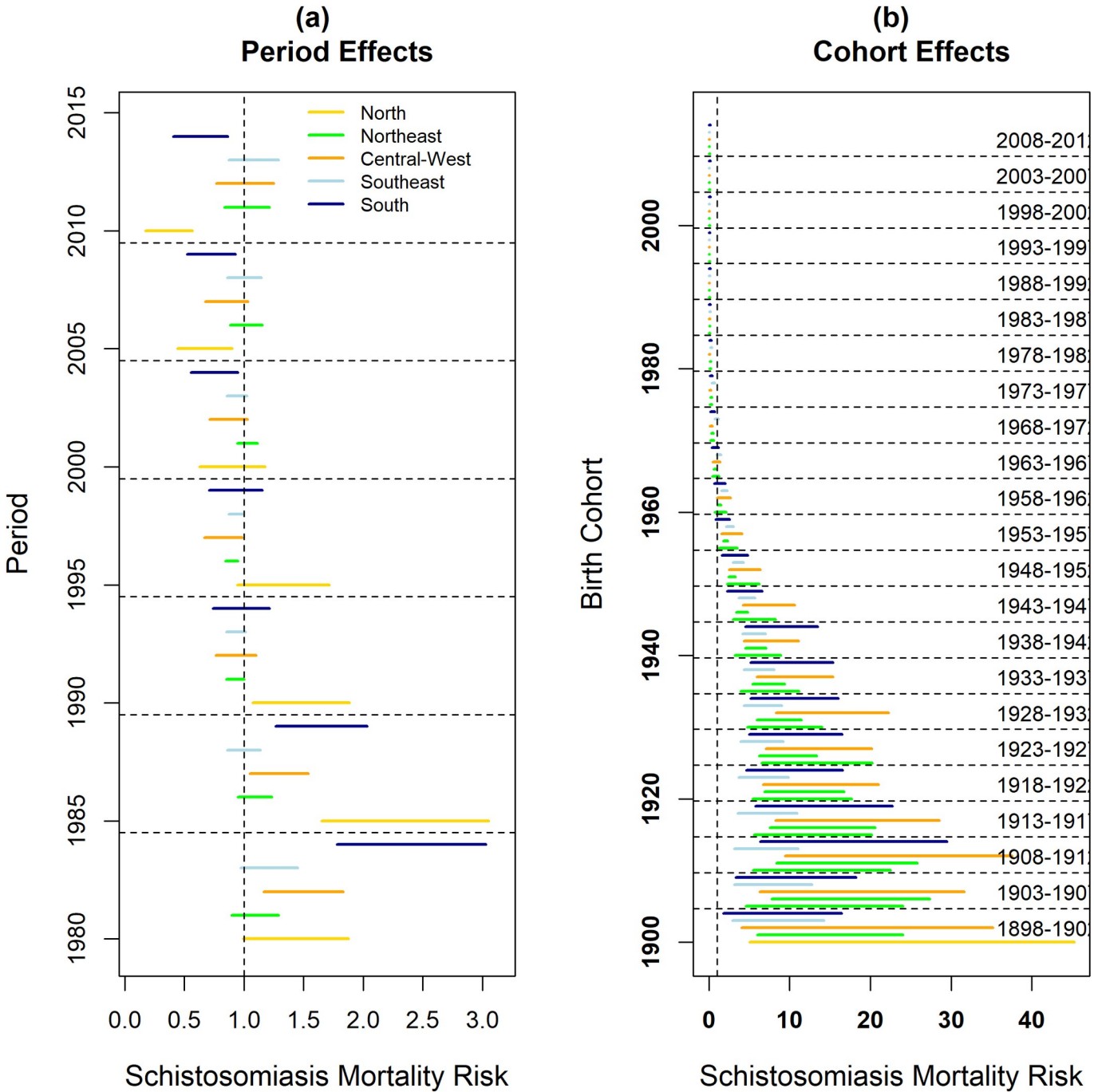

**Fig 4.** 95% Credibility intervals for the estimated risks of death from schistosomiasis, according to period (a) and birth cohort (b), and according to geographic regions, Brazil, 1980–2014.

The reduction in mortality from schistosomiasis in important endemic regions in the country can be partially attributed to the impact of the control program measures implemented in the last decades, based mainly on the large-scale treatment of the population [7,9,29]. The Special Program for the Control of Schistosomiasis (PECE), developed by the Ministry of Health, started in 1976, in eight states in the Northeast of Brazil, and aimed to reduce the prevalence of

*S. mansoni* infection to 4% [29]. From 1990 to 2002, the National Health Foundation (Funasa) started to run the Schistosomiasis Control Program (PCE), and in 1994, professionals in Primary Care / Family Health and Health Surveillance began to act in the control of schistosomiasis in an integrated way [1].

In 2011, the Ministry of Health launched the Integrated Strategic Action Plan for the Elimination of Leprosy, Filariasis, Schistosomiasis and Onchocerciasis as Public Health Problems, Trachoma as a cause of blindness and control of geohelminthiasis, assuming the political commitment to face these diseases prevalent in the most vulnerable groups of Brazilians [33]. Casavechia et al. (2018), through systematic review and meta-analysis, showed a prevalence greater than 18% for *S. mansoni*. These findings suggested a positive response to control programs, although with mixed results in the country. A reduction in disease transmission was observed in all studies that examined prevalence after the implementation of monitoring programs. However, although there was a reduction, the disease transmission remained active in all the analyzed sites [34].

Martins-Melo et al. (2014) also observed a higher mortality rate in the northeast region, with greater transmission on the coast and forest areas, where ecological conditions and the organization of the space provide intense transmission. In addition, the transposition of the São Francisco River, the largest river in the region, may have contributed to the emergence of diseases, dispersing intermediate hosts in previously non-endemic areas, and increasing the migratory activities of construction workers and their families [35].

Silveira et al. (1990) showed a reduction in mortality between 1977 and 1986, especially in the southeastern region of Brazil, due to the decrease in the number of severe cases of the disease due to new drugs, administered in a single dose and on a large scale. The treated and untreated populations that remained in the Northeast were subject to reinfection, while the risk was greatly reduced or eliminated for the population that migrated to the Southeast and other areas. The majority of deaths attributed to schistosomiasis in the Southeast region occurred among northeastern migrants, especially those living in the state of São Paulo. Another fact that may have influenced the greatest reduction in schistosomiasis mortality in the Southeast is the largest and supposedly better medical care, with the use of specific medications by the permanent health services [7].

As for the North region, Martins-Melo et al. (2015) shown an increasing number of confirmed cases in recent years in the state of Rondônia, where the majority of cases and deaths were not autochthonous, but were identified in migrants from regions endemic to schistosomiasis in Brazil. The presence of potential intermediate hosts was confirmed in Rondônia, increasing the possibility of the disease being established [35].

The higher risk of death from schistosomiasis in older age groups can be explained by the chronic nature of the disease, progressing to severe clinical forms and the occurrence of more frequent chronic comorbidities in the elderly [8,30]. Firmo et al. (1996), in a cross-sectional study carried out in a suburban area of a large industrialized city in Brazil, observed that age was the variable with the highest association with *S. mansoni* infection, with the highest Odds Ratio in the second decade of life [36]. Some studies have observed that the reduction in mortality over time has been more significant in age groups under 30 years. This can be attributed to the fact that control programs based on diagnosis and selective chemotherapy treatments are generally focused on children and adolescents (school-age children) in endemic areas [7,8,30,38].

No studies found evaluating the association of death from schistosomiasis and sex. However, some studies indicate that the association between sex and the risk of infection for schistosomiasis is ambiguous and culturally variable. Firmo et al. (1996), in relation to the prevalence of infection in males, claim that it is probably a consequence of different patterns of

contact with water between men and women [36]. The study by Dalton and Pole in the Volta Lake region (1978) demonstrated the extent to which sex was predictive of infection, related with domestic activities in contact with water and activities associated with fishing and canoeing, including their use in economic and recreational activities [37,39]. In addition, religion and women's mobility restrictions imposed by some beliefs and religion, are one of these influences [8,30,38–40].

In this study, it was found that people born between 1903 and 1912 had a higher risk of mortality from schistosomiasis. Protective effect was seen in people born after 1968. The effects of the birth cohort on mortality differed with geographic regions. In fact, most deaths can be considered to be related to cases of past infections, considering the average age of the deaths and the natural history of the disease [9]. Protective immune responses against schistosomiasis develop slowly over a period of 10 to 15 years, and children under 10 years of age, in areas endemic to schistosomiasis, are susceptible to reinfection after treatment, while adults are generally resistant [41]. Other factors, in addition to those related to the control programs, also contributed to the reduction of schistosomiasis mortality among the youngest cohorts in Brazil, such as increased urbanization, general improvements in socioeconomic and sanitary conditions, and improved health services. Schistosomiasis has expanded widely across the country due to migratory movements to areas with poor sanitation [1,7,9,30,38].

As limitations of this study, it can be mentioned that the number of deaths due to schistosomiasis is difficult to estimate due to hidden pathologies, such as liver and kidney failure, bladder cancer and ectopic pregnancy due to female genital schistosomiasis. Another counterpoint of this work is the quality of information and the coverage of the mortality information system. Although we performed the correction of death records for ill-defined causes, in addition to ignored sex and age, we did not correct the underreporting of deaths and, therefore, schistosomiasis mortality rates may be underestimated [7,9].

This study showed the important role of younger birth cohorts in the overall reduction of schistosomiasis mortality over time in Brazil and geographic regions. This effect reflects the effectiveness of disease control programs and improvements in access to health and socioeconomic conditions of the population over time. The effects of age, period and cohort also varied according to sex and geographic regions, corroborating the different habits and living conditions of the Brazilian population.

## Author Contributions

**Conceptualization:** Taynãna César Simões, Roberto Sena, Karina Cardoso Meira.

**Data curation:** Taynãna César Simões, Karina Cardoso Meira.

**Formal analysis:** Taynãna César Simões, Karina Cardoso Meira.

**Investigation:** Taynãna César Simões, Karina Cardoso Meira.

**Methodology:** Taynãna César Simões, Karina Cardoso Meira.

**Project administration:** Taynãna César Simões.

**Supervision:** Taynãna César Simões.

**Validation:** Taynãna César Simões, Karina Cardoso Meira.

**Visualization:** Taynãna César Simões, Karina Cardoso Meira.

**Writing – original draft:** Taynãna César Simões, Karina Cardoso Meira.

**Writing – review & editing:** Taynãna César Simões, Roberto Sena, Karina Cardoso Meira.

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
