## [Decision Letter · Decision Letter 0]

28 Jan 2020

PONE-D-19-22417

The influence of the cohort effect on the variability of schistosomiasis mortality trend in Brazil

PLOS ONE

Dear PhD Simões,

Thank you for submitting your manuscript to PLOS ONE. After careful consideration, we feel that it has merit but does not fully meet PLOS ONE’s publication criteria as it currently stands. Therefore, we invite you to submit a revised version of the manuscript that addresses the specific points raised during the review process.

We would appreciate receiving your revised manuscript by Mar 13 2020 11:59PM. To enhance the reproducibility of your results, we recommend that if applicable you deposit your laboratory protocols in protocols.io, where a protocol can be assigned its own identifier (DOI) such that it can be cited independently in the future. For instructions see: http://journals.plos.org/plosone/s/submission-guidelines#loc-laboratory-protocols

We look forward to receiving your revised manuscript.

Kind regards,

David Joseph Diemert, M.D.

Academic Editor

PLOS ONE

Journal Requirements:

2. Please amend the subsection category “[FOR JOURNAL STAFF USE ONLY]” for your manuscript. Unfortunately, this is not a valid category. At this time, please choose one or more subsections that best represent the topic(s) of your study.

4. Please include your tables as part of your main manuscript and remove the individual files. Please note that supplementary tables (should remain/ be uploaded) as separate "supporting information" files

Reviewers' comments:

Reviewer's Responses to Questions

**Comments to the Author**

1. Is the manuscript technically sound, and do the data support the conclusions?

Reviewer #1: Yes

Reviewer #2: No

2. Has the statistical analysis been performed appropriately and rigorously? 

Reviewer #1: I Don't Know

Reviewer #2: Yes

3. Have the authors made all data underlying the findings in their manuscript fully available?

Reviewer #1: Yes

Reviewer #2: Yes

4. Is the manuscript presented in an intelligible fashion and written in standard English?

Reviewer #1: No

Reviewer #2: No

5. Review Comments to the Author

Reviewer #1: The theme of the manuscript is important regarding mortality and population infected with Schistosoma mansoni in Brazil. It also compares different age groups, as well as different populations from the socioeconomic point of view. However the English is poor and must be improved.

Reviewer #2: This study examined the effects of age, period and birth cohort (APC Effects) on the temporal evolution of schistosomiasis mortality in Brazil, according to sex and the five Brazilian geographic regions from 1980 to 2014. The results show that about 20,000 schistosomiasis deaths were recorded in Brazil during the study period. The highest mortality rates from schistosomiasis occurred at the most advanced ages, in men and in the Northeast region. The mortality rates increased with increasing age, declined over the years, and were higher for older generations or birth cohorts.

The study has great relevance in the area. However, there are important issues/points that need to be improved in the manuscript before being reconsidered:

-I suggest a comprehensive review of the English written in the manuscript (typographical / grammatical errors and scientific writing);

Title:

-Lines 1-2: Please identify the time period (1980-2014) and replace “cohort effects” by “age-period-cohort effects”;

Abstract:

-Background: Please include the overall objective of the study;

-Methods: Please identify the study design; Are the population data extracted from the SIM or IBGE?

-Results: The description of the findings is confusing and difficult to understand; -Please replace "Midwest" with "Central-West" in the entire manuscript (e.g., Line 40);

-The conclusions should be focused/based on the key findings of the study.

Author summary:

-Most of this section is based on the description of the background, with an insufficient description of the main findings of the study.

Introduction:

-This section is relatively long and could be shortened focusing on contextualization and justification of the study problem.

Methods:

-Lines 146-156: Please include a more updated estimate of the Brazilian population (e.g., 210.1 million in 2019);

-Lines 157-181: Please include more information and details of the national databases used (e.g. SIM database, death certificate [DO] as standard death registration document, etc.) and explain and clarify how mortality and population data were extracted from national databases (downloading the SIM microdata or data tabulation through the DATASUS website); In addition, please specify the period/year of data collection and clarify whether schistosomiasis deaths were extracted as a underlying or multiple causes of death (underlying and associated causes of death).

Results:

-Please replace "basic cause" with "underlying cause" in the entire manuscript (e.g., Line 229);

-Please stratify the description of the findings in more subsections.

-Line 233: Is the value of this mortality rate correct (2.99 deaths/100,000 inhabitants)?

-Discussion:

-The discussion is long and somewhat repetitive in some parts, requiring a comprehensive review. I recommend that the authors show each important point found in the study in a logical sequence according to the Results section, immediately before discussing possible reasons found in the literature. It would be clearer if each paragraph represents a single idea. In addition, please do not repeat the findings described in the Results section;

-Limitations: Please include the limitations on the quality and coverage of mortality data in Brazil over the study period;

-Please finalize the Discussion section with the conclusion(s) of the study.

6. PLOS authors have the option to publish the peer review history of their article (what does this mean?). If published, this will include your full peer review and any attached files.

Reviewer #1: No

Reviewer #2: No

---

## [Author Response · Author response to Decision Letter 0]

27 Feb 2020

Reviewer #1: The theme of the manuscript is important regarding mortality and population infected with Schistosoma mansoni in Brazil. It also compares different age groups, as well as different populations from the socioeconomic point of view. However the English is poor and must be improved.

Answer:

We are grateful for the reviewer's contribution and we certify that we have reviewed the entire writing of the manuscript, as well as the adequacy of the language.

Reviewer #2: This study examined the effects of age, period and birth cohort (APC Effects) on the temporal evolution of schistosomiasis mortality in Brazil, according to sex and the five Brazilian geographic regions from 1980 to 2014. The results show that about 20,000 schistosomiasis deaths were recorded in Brazil during the study period. The highest mortality rates from schistosomiasis occurred at the most advanced ages, in men and in the Northeast region. The mortality rates increased with increasing age, declined over the years, and were higher for older generations or birth cohorts.

The study has great relevance in the area. However, there are important issues/points that need to be improved in the manuscript before being reconsidered:

-I suggest a comprehensive review of the English written in the manuscript (typographical / grammatical errors and scientific writing);

Answer:

We are grateful for the reviewer's contribution and we certify that we have reviewed the entire writing of the manuscript, as well as the adequacy of the language.

Title:

-Lines 1-2: Please identify the time period (1980-2014) and replace “cohort effects” by “age-period-cohort effects”;

Answer: 

We changed the title to: “The influence of the age-period-cohort effects on the temporal trend of mortality from schistosomiasis in Brazil from 1980 to 2014”.

Abstract:

-Background: Please include the overall objective of the study;

Answer:

We added this sentence to the summary “In this study, the temporal change in mortality was evaluated in relation to the effects of age, period and birth cohort, in Brazil and regions, from 1980 to 2014”.

-Methods: Please identify the study design; Are the population data extracted from the SIM or IBGE?

Answer:

We completed the first sentence of the methods section: “This is a population-based longitudinal ecological study to analyze the time trend of schistosomiasis mortality rates in Brazil” (page 6, line 137). In addition, we clarify where population data were obtained: “Population data were obtained from the IBGE website (Brazilian Institute of Geography and Statistics) [16]” (page 7, line 164). 

-Results: The description of the findings is confusing and difficult to understand; -Please replace "Midwest" with "Central-West" in the entire manuscript (e.g., Line 40);

Answer:

We reviewed and rewrote part of the results section. In addition, we made the suggested replacement for "Central-West".

-The conclusions should be focused/based on the key findings of the study.

Answer:

We reviewed and rewrote the conclusions:

“The birth cohorts had a great influence on the decreasing trend of schistosomiasis mortality in Brazil. This result may be due to the interaction between demographic changes and greater access to health and sanitation services, in addition to the impact of schistosomiasis control measures experienced by younger cohorts.”

Author summary:

-Most of this section is based on the description of the background, with an insufficient description of the main findings of the study.

Answer:

We rewrote this section with a focus on the methodology used and the results obtained:

“Schistosomiasis infection is prevalent in tropical and subtropical areas in poor communities, without access to drinking water and without adequate sanitation, affecting millions of people worldwide. In Brazil, Schistosomiasis mansoni is endemic in a large part of the territory and considered an important public health problem. In this study, we analyzed the influence of age, period and birth cohort (APC effects) on the temporal evolution of schistosomiasis mortality in Brazil and its geographic regions, between 1980 and 2014. The APC effects varied according to sex and regions. Men were at risk of dying between 25 and 54 years of age, while risk for women occurred after seventy years of age. Birth Cohorts had the greatest influence on the trend of decreasing mortality, with a protective effect on death among people born after 1973. Demographic changes and greater access to health experienced by younger generations may have a role important in decreasing deaths from the disease.”

Introduction:

-This section is relatively long and could be shortened focusing on contextualization and justification of the study problem.

Answer:

We have reviewed and summarized this section, focusing on the problems highlighted by the reviewer.

Methods:

-Lines 146-156: Please include a more updated estimate of the Brazilian population (e.g., 210.1 million in 2019);

Answer:

We updated population estimates for 2019: 

“This is a population-based longitudinal ecological study to analyze the time trend of schistosomiasis mortality rates in Brazil. The country is divided into 26 states and a federal government grouped into five major regions (North, Northeast, Southeast, South and Central-West) with different geographical, economic and cultural characteristics. The Southeast (88,072,407 inhabitants) is the most populous and economically most developed region, and the South region (30,036,030 inhabitants) has the best human development indexes. On the other hand, the Central-West (16,293,774 inhabitants) has an economy based on agriculture and livestock, the Northeast (57,883,049 inhabitants) has the lowest human development rates and the North (18,373,753 inhabitants), which includes the Amazon rainforest, has low population density and ranks second with the lowest human development rates. Population data were obtained from the IBGE website (Brazilian Institute of Geography and Statistics).”

-Lines 157-181: Please include more information and details of the national databases used (e.g. SIM database, death certificate [DO] as standard death registration document, etc.) and explain and clarify how mortality and population data were extracted from national databases (downloading the SIM microdata or data tabulation through the DATASUS website); In addition, please specify the period/year of data collection and clarify whether schistosomiasis deaths were extracted as a underlying or multiple causes of death (underlying and associated causes of death).

Answer:

We rewrote the sentence related to obtaining the data we used, making it clearer and more detailed. In addition, we added the collection period and the type of death analyzed (underlying death).

Page 7, lines 149-162: “The annual death records between 1980 and, specific for age, were extracted from the Mortality Information System of the SUS (Unified Health System) Department of Information Technology (SIM-DATASUS) of the Ministry of Health of Brazil (DATASUS, 2017). The categories of underlying deaths selected in the SIM were Schistosomiasis - codes 120 (ICD-9 until 1995) and B65 (ICD-10 after 1996). SIM was created for the regular collection of data on mortality in the country, allowing the collection of data in a comprehensive way, in order to subsidize the different spheres of management in public health. Death certificates contain demographic (age, gender, education, race, marital status, municipality of residence and municipality of occurrence of death) and clinical information (underlying and associated causes of death). It is the physicians’ responsibility to complete the death certificates. Until 1995, reference codes were based on the International Classification of Diseases (ICD) in its 9th revision, and after 1996, they were based on the 10th revision of the ICD. All data were collected in November 2017 [13].” 

Page 7, lines 147-48: “Population data were obtained from the IBGE website (Brazilian Institute of Geography and Statistics). 

Results:

-Please replace "basic cause" with "underlying cause" in the entire manuscript (e.g., Line 229);

Answer:

We made the suggested replacement.

-Please stratify the description of the findings in more subsections.

Answer: 

We stratified the description of the findings in the subsections: "Describing the data", "APC effects on mortality in Brazil" and "APC effects on mortality in regions".

-Line 233: Is the value of this mortality rate correct (2.99 deaths/100,000 inhabitants)?

Answer:

The value is correct, but it is the crude rate calculated for the entire period of analysis. We rewrote the sentence.

Page 10, lines 233-35: “The crude death rate from schistosomiasis was 2.99 deaths per 100,000 inhabitants during the entire period analyzed (1980-2014). The standardized annual average rate was 0.45 deaths per 100,000 inhabitants.”

-Discussion:

-The discussion is long and somewhat repetitive in some parts, requiring a comprehensive review. I recommend that the authors show each important point found in the study in a logical sequence according to the Results section, immediately before discussing possible reasons found in the literature. It would be clearer if each paragraph represents a single idea. In addition, please do not repeat the findings described in the Results section;

Answer:

We have reviewed and summarized this section, focusing on the issues highlighted by the reviewer.

-Limitations: Please include the limitations on the quality and coverage of mortality data in Brazil over the study period;

Answer:

We included a sentence about the quality and coverage of the data as limitations of the study.

Page 15, lines 411-18: “As limitations of this study, it can be mentioned that the number of deaths due to schistosomiasis is difficult to estimate due to hidden pathologies, such as liver and kidney failure, bladder cancer and ectopic pregnancy due to female genital schistosomiasis. Another counterpoint of this work is the quality of information and the coverage of the mortality information system. Although we performed the correction of death records for ill-defined causes, in addition to ignored sex and age, we did not correct the underreporting of deaths and, therefore, schistosomiasis mortality rates may be underestimated [7,9].”

-Please finalize the Discussion section with the conclusion(s) of the study.

Answer:

We conclude the Discussion section with a paragraph of conclusions.

Page 15, lines 419-24: “This study showed the important role of younger birth cohorts in the overall reduction of schistosomiasis mortality over time in Brazil and geographic regions. This effect reflects the effectiveness of disease control programs and improvements in access to health and socioeconomic conditions of the population over time. The effects of age, period and cohort also varied according to sex and geographic regions, corroborating the different habits and living conditions of the Brazilian population.”

---

## [Decision Letter · Decision Letter 1]

3 Apr 2020

The influence of the age-period-cohort effects on the temporal trend mortality from schistosomiasis in Brazil from 1980 to 2014

PONE-D-19-22417R1

Dear Dr. Simões,

We are pleased to inform you that your manuscript has been judged scientifically suitable for publication and will be formally accepted for publication once it complies with all outstanding technical requirements.

With kind regards,

David Joseph Diemert, M.D.

Academic Editor

PLOS ONE

Additional Editor Comments (optional):

Reviewers' comments:

Reviewer's Responses to Questions

**Comments to the Author**

1. If the authors have adequately addressed your comments raised in a previous round of review and you feel that this manuscript is now acceptable for publication, you may indicate that here to bypass the “Comments to the Author” section, enter your conflict of interest statement in the “Confidential to Editor” section, and submit your "Accept" recommendation.

Reviewer #1: All comments have been addressed

Reviewer #2: All comments have been addressed

2. Is the manuscript technically sound, and do the data support the conclusions?

Reviewer #1: Yes

Reviewer #2: Partly

3. Has the statistical analysis been performed appropriately and rigorously? 

Reviewer #1: I Don't Know

Reviewer #2: Yes

4. Have the authors made all data underlying the findings in their manuscript fully available?

Reviewer #1: Yes

Reviewer #2: Yes

5. Is the manuscript presented in an intelligible fashion and written in standard English?

Reviewer #1: Yes

Reviewer #2: Yes

6. Review Comments to the Author

Reviewer #1: The subject under study in this manuscript is of relevant importance with regard to the death of the population infected with Schistosoma mansoni in Brazil. The study is candated careful making a comparison with different age groups, as well as populations with a different socioeconomic status. Finally, they did the study over a wide period of time (1980-2014).

Reviewer #2: (No Response)

7. PLOS authors have the option to publish the peer review history of their article (what does this mean?). If published, this will include your full peer review and any attached files.

Reviewer #1: No

Reviewer #2: No

---

## [Editor Report · Acceptance letter]

9 Apr 2020

PONE-D-19-22417R1 

The influence of the age-period-cohort effects on the temporal trend mortality from schistosomiasis in Brazil from 1980 to 2014 

Dear Dr. Simões:

I am pleased to inform you that your manuscript has been deemed suitable for publication in PLOS ONE. Congratulations! Your manuscript is now with our production department. 

With kind regards,

on behalf of

Dr. David Joseph Diemert 

Academic Editor

PLOS ONE